# Mechanisms of *Salmonella typhimurium* Resistance to Cannabidiol

**DOI:** 10.3390/microorganisms13030551

**Published:** 2025-02-28

**Authors:** Iddrisu Ibrahim, Joseph Atia Ayariga, Junhuan Xu, Daniel A. Abugri, Robertson K. Boakai, Olufemi S. Ajayi

**Affiliations:** 1The Microbiology Program, College of Science, Technology, Engineering, and Mathematics, Alabama State University, Montgomery, AL 36104, USA; iiddrisu3715@alasu.edu (I.I.); dabugri@alasu.edu (D.A.A.); brobertson@alasu.edu (R.K.B.); 2The Industrial Hemp Program, College of Science, Technology, Engineering, and Mathematics, Alabama State University, Montgomery, AL 36104, USA; jxu@alasu.edu

**Keywords:** antimicrobials, resistance, Cannabidiol, *Salmonella*, *blaTEM*, *fimA*, lipopolysaccharide, ergosterols

## Abstract

The emergence of multi-drug resistance (MDR) poses a huge risk to public health globally. Yet these recalcitrant pathogens continue to rise in incidence rate with resistance rates significantly outpacing the speed of antibiotic development. This therefore presents related health issues such as untreatable nosocomial infections arising from organ transplants and surgeries, as well as community-acquired infections that are related to people with compromised immunity, e.g., diabetic and HIV patients, etc. There is a global effort to fight MRD pathogens spearheaded by the World Health Organization, thus calling for research into novel antimicrobial agents to fight multiple drug resistance. Previously, our laboratory demonstrated that Cannabidiol (CBD) is an effective antimicrobial against *Salmonella typhimurium (S. typhimurium*). However, we observed resistance development over time. To understand the mechanisms *S. typhimurium* uses to develop resistance to CBD, we studied the abundance of bacteria lipopolysaccharide (LPS) and membrane sterols of both CBD-susceptible and CBD-resistant *S. typhimurium* strains. Using real-time quantitative polymerase chain reaction (rt qPCR), we also analyzed the expression of selected genes known for aiding resistance development in *S. typhimurium*. We found a significantly higher expression of *blaTEM* (over 150 mRNA expression) representing over 55% of all the genes considered in the study, *fimA* (over 12 mRNA expression), *fimZ* (over 55 mRNA expression), and *integron 2* (over 1.5 mRNA expression) in the CBD-resistant bacteria, and these were also accompanied by a shift in abundance in cell surface molecules such as LPS at 1.76 nm, ergosterols at 1.03 nm, oleic acid at 0.10 nm and MPPSE at 2.25nm. For the first time, we demonstrated that CBD-resistance development in *S. typhimurium* might be caused by several structural and genetic factors. These structural factors demonstrated here include LPS and cell membrane sterols, which showed significant differences in abundances on the bacterial cell surfaces between the CBD-resistant and CBD-susceptible strains of *S. typhimurium*. Specific key genetic elements implicated for the resistance development investigated included *fimA*, *fimZ*, *int2*, *ompC*, *blaTEM*, DNA recombinase (*STM0716*), leucine-responsive transcriptional regulator (*lrp/STM0959*), and the *spy* gene of *S. typhimurium*. In this study, we revealed that *blaTEM* might be the highest contributor to CBD-resistance, indicating the potential gene to target in developing agents against CBD-resistant *S. typhimurium* strains.

## 1. Introduction

Antimicrobial resistance has successfully become a global health menace, and resistance is often acquired by bacteria through health-care-related incidence (HRI) orchestrated by multiple drug-resistant (MDR) and extended drug-resistant pathogens (EDRP) such as *Klebsiella pneumonia*, *Staphylococcus aureus*, *Enterococcus faecium*, *Acinetobacter baumannii*, *Enterobacter* spp., and *Pseudomonas aeruginosa* [1,2,3,4]. The recalcitrance of pathogenic bacteria indicates that millions of people are at risk of infection [5,6,7]. This has forced the Centre for Disease Control and Prevention (CDC) to change its 2019 memorandum from “a coming post-antibiotic epoch” to the fact that we are already in the era of stringent antibiotic resistance [7].

Understanding the mechanisms of antimicrobial resistance is a baffling task that challenges both scientific intellect and economics [3,8], and several pharmaceutical companies have given up on antibiotic research. The perilous financial situation of antibiotic research and development means that there are fewer novel therapeutics underway, and not even a single novel antibiotic class has been developed and approved for clinical application since the 1960s, especially for Gram-negative bacteria [3,8,9].

Salmonellosis (an infection caused by *Salmonella*, often through contaminated food or water) is lethal, and approximately 1.35 million infections, 26,500 hospitalizations, and 420 mortality cases are reported annually in the United States (US) alone [10,11]. Stomach cramps, diarrhea, and fever are some common symptoms of salmonellosis, and these symptoms develop based on the six-six principle (6 h-6 days), which may last four to seven days in symptomatic people. Some people may be asymptomatic or may only show symptoms after several weeks of infection [12,13]. Salmonellosis can be prevented when hygiene is strictly practiced, such as eating foods that are pasteurized, regular hand washing, keeping kitchens and restrooms clean, etc. [14,15]. However, like all other pathogens, *Salmonella* is capable of escaping hygienic protocols and developing resistance to most antibiotics [16].

To curtail the resistance of pathogenic bacteria, CBD, the primary anti-psychoactive component of the hemp plant, has increasingly been studied and employed. It is a small molecule with a molecular weight of 314 Da and chemical formulae C_2__1_H_3__0_O_2_, consisting mainly of a pentyl-substituted bis-phenol aromatic class [17,18]. These aromatic classes are often linked to the alkyl-substituted cyclohexane terpene ring system [19,20]. Among the dozens of cannabinoids extracted from the hemp plant, CBD is one of them and has proven to possess some bioactive components that act as antimicrobials [21,22]. Though CBD was first isolated in 1940, it was not until 1963 that the full understanding of its structure was unveiled [23,24]. Since the elucidation of its structure, CBD has been rigorously tested and applied as an antidote to several diseases such as spasticity due to multiple sclerosis, depression, appetite stimulation, sleep disorders, glaucoma, psychosis, and anxiety disorders, among many others. Another pharmacological relevance of CBD is its potential application in neuroprotective and anti-inflammatory diseases [25]. Epidiolex® in the US and Epidyolex in the European Union are so far the oil-based liquid formulations of CBD that have been approved, respectively, by the US Food and Drug Administration in 2018 and by the European Medicines Agency in 2019. These formulations were critically evaluated and shown to play a role in the oral treatment of epilepsy disorders such as Lennox-Gastaut and Dravet syndrome [26,27]. CBDs antimicrobial ability has been widely reported and shown to have minimum inhibitory concentrations (MICs) in the range of 1–5 μg/mL over Gram-positive bacteria, particularly *Streptococci* and *Staphylococci*, and Gram-negative bacteria such as *Salmonella* [28].

Notwithstanding the rich applications of CBD as an antimicrobial, most bacteria have developed strategies of resisting CBD similar to most antimicrobials, and this poses a threat to public health [29,30]. Recently we have demonstrated that CBD has potent antimicrobial activity against two strains of *Salmonella* (*S. typhimurium and S.* Newington), as compared to some common broad-spectrum antibiotics such as polymyxin B, Azitromycin, and Kanamycin [28,31]. However, we observed over time that *Salmonella* developed resistance against CBD. Given this, this present study seeks to elucidate the mechanisms that *S. typhimurium* employs to develop resistance to CBD.

## 2. Materials and Methods

### 2.1. Media, Chemicals, Bacterial Strains, and Other Reagents

All chemicals (Hexane, methanol–LC-MS (≥99.9%), water, and ethanol absolute proof (≥99.5%)) used in this research were of HPLC grade. Dulbecco’s Modified Eagle Medium (DMEM) and Fetal Bovine Serum (FBS) were purchased from ATCC (Manassa, CO, USA). Our laboratory has demonstrated that *S. typhimurium* is susceptible to CBD at micromolar concentrations [28]. The CBD used in this study extraction and purification process was carried out by Sustainable CBD LLC. (Salem, AL, USA) and has been described elsewhere [28]. The PCR ribotype of the *S. typhimurium* LT2 strain MS1868 used in the study was named BV4012 strains, which is the CBD-susceptible strain (a kind gift from Dr. Anthony R. Poteete, University of Massachusetts). The second PCR ribotype of *S. typhimurium* used in this study was the CBD-resistant strain created from BV4012 (To create CBD-resistant colonies, the CBD-susceptible *S*. *typhimurium* strains were subjected to prolonged CBD pressure). The resistant colonies were picked and plated on Luria Broth (BD, Difco, Franklin Lakes, NJ, USA) and Luria Agar (BD, Difco, Franklin Lakes, NJ, USA) premixed with CBD at 10 μg/mL final concentration. From hence all cultures of CBD-resistant strains received 10 μg/mL CBD concentration or more in their growth media. The Vero cells (ATCC, CCL 81) used in this study were obtained from BEI Resources (Manassas, VA, USA). The Vero cells were cultured and maintained at 37 °C, 5% CO_2_ in a T-25 cm3 flask using DMEM supplemented with 10% FBS and 1% penicillin-streptomycin-amphotericin [32].

### 2.2. Extraction of Lipids

The CBD-resistant strain has been previously developed in our lab [31]. CBD-resistant or CBD-susceptible *S. typhimurium* cells were cultured to mid-log stage in LB broth only for the susceptible or CBD-tinted LB broth for the CBD-resistant strain, then 10 mL of the culture was pelleted and resuspended with an equal volume of the LB broth only or LB broth tinted with CBD with a final concentration of (0.01 mg/mL). The bacteria cultures were allowed to grow at room temperature for 6 h. The LB broth was removed after centrifugation, and the pellet was washed with 1X Phosphate-Buffered Saline (PBS) twice. The pellet was then resuspended in 5 mL 1X PBS and aliquoted into 2 mL Eppendorf tubes at 1 mL volume per tube. Bacterial lipids were extracted from the cells using the Bligh/Dyer procedure [33] with fewer modifications. Thus, 500 μL of chloroform:methanol (1:2 *v*/*v*) was added to the resuspended cells in the 2 mL Eppendorf tubes. The samples were then vortexed at high speed for 20 s each and then centrifuged at 529× *g* for 5 min to pellet insoluble cellular debris. Then, 500 μL of the resulting supernatant was decanted into new 2 mL Eppendorf tubes, and 500 μL each of chloroform and PBS was added. The samples were vortexed for the second time for 10 s and centrifuged at room temperature for 5 min at 529× *g*. The resulting two liquid phases were observed, and then the organic lower phase was carefully transferred into a fresh 2 mL Eppendorf tube using sterile pipette tips. Then, the upper aqueous phase was discarded.

### 2.3. Ergosterols Quantification Using UV-Vis Spectrophotometry

Sterols with conjugated double bonds, for example, ergosterol, have been demonstrated to have strong absorbances between 250 and 300 nm and have the capacity at this wavelength to detect at a limit of 6 ng of ergosterol. It has been demonstrated that ergosterols peak at 295 nm and have a shoulder at 265 nm [34]. The amount of ergosterols in CBD-treated *S*. *typhimurium* samples or the untreated controls was quantified using the ABS Spectral Max UV-Vis spectrophotometer (Molecular Devices LLC., San Jose, CA, USA). Approximately 500 μL of the extracted lipid was aliquoted into 1 mL cuvette tubes, and the absorbance was measured at 295 nm. Three replicate measurements were made for each treatment, and the absorbance was averaged.

### 2.4. Mysristic, Palmitic Acid, Palmitoleic Acid, Stearic Acid, Erucic Acid, and Oleic Acids Quantification Using UV-Vis Spectrophotometry

Mysristic, palmitic acid, palmitoleic acid, stearic acid, and erucic acid were classified in this work as a single group of sterols since they all have their absorbance at 208 nm [35]. Thus, this group of sterol quantifications was made via measuring the absorbance of the extracted lipids at 208 nm. Oleic acids, however, are known to have absorbance at 330 nm [36]; thus, the quantification of oleic acid was carried out via measuring absorbance at 330 nm.

### 2.5. Unsaturated and Other Uncategorized Bacterial Membrane Sterols Quantification Using UV-Vis Spectrophotometry

#### LPS Extraction Protocol

To compare the relative quantities of unsaturated and saturated bacterial membrane sterols, membrane unsaturated fatty acids were extracted using the Lipid Assay Kit (ab242305 Abcam, Waltham, MA, USA); the lipids were aliquoted into 1 mL cuvette tubes and were measured at an absorbance of 540 nm using the UV-Vis spectrophotometer. For all other uncategorized membrane sterols, absorbance at 314 nm was used to quantify them.

### 2.6. Quantitative PCR Analysis of Gene Expression

To investigate the effect of CBD on the antibiotic resistance genes in *S. typhimurium*, the total RNA of CBD-resistant and CBD-susceptible *S. typhimurium* strains (n = 3) was extracted using the RNeasy kit (Qiagen Sciences, Germantown, MD, USA) after overnight culture in CBD-tinted LB broth (for the CBD-resistant strain) and LB broth only (for the CBD-susceptible strains). After extraction, the RNA was dissolved in RNase-free water, and the concentration was determined by Nanodrop (1000 spectrophotometer, ThermoFisher Scientific, Madison, WI, USA). Following the RNA extraction, 100 ng of RNA from each sample was then reverse transcribed into cDNA using the SSIV Cell Direct cDNA Synthesis kit (lot# 00916637, Invitrogen, ThermoFisher Scientific, Vilnius, Lithuania). Subsequently, the cDNA was subjected to quantitative PCR analysis for the following antibiotic markers: *int1*, *int2*, *int3*, *blaTEM*, *fimA*, *fimZ*, *STM0959*, *STM0716*, and the *spy* genes, using CFX96TM real-time PCR (BioRad Laboratories, Hercules, CA, USA) following similar procedures as in our previous work [37].

The primers for *int1*, *int2*, *int3*, *blaTEM*, *fimA*, *fimZ*, *STM0959*, *STM0716*, and the *spy* genes were designed and ordered from ThermoFisher Scientific, and the DNA gyrase B (*gyrB*) gene served as the housekeeping gene. Real-time QPCR analysis of *int1*, *int2*, *int3*, *blaTEM*, *fimA*, *fimZ*, *STM0959*, *STM0716*, and the *spy* genes and the *S. typhimurium* housekeeping gene (*gyrB*) was carried out using the primers listed in Table 1 and PowerUp™ SYBR™ Green Master Mix (lot# 00914521). Comparisons of target gene expression in the samples were carried out using the cycle threshold (Ct) normalized to the housekeeping gene *gyrB*.

### 2.7. Anti-Invasion Assay

To assess the ability of CBD-resistant *S. typhimurium* to resist the CBD as a prophylactic agent in Vero cell lines, Vero cells at a density of 1 × 10^5^ were cultured at 37 °C in an atmosphere containing 5% CO_2_ in DMEM media supplemented with 10% Fetal Bovine Serum (FBS) (Gibco, Grand Island, NY, USA) in a 96-tissue culture plate (Corning Costar, Milano, Italy) for 24 h. Then, the media was removed, and 300 μL of CBD-resistant or CBD-susceptible *S. typhimurium* cells at an OD_600_ of 0.3 were added to the growing Vero cells. This was then immediately followed with the administration of CBD at 100 μg/mL, or media control supplemented DMEM media only. Infection was allowed for 30 min following a similar procedure by Ayariga et al., and Manini et al. [32,41]. The final concentrations of CBD in the culture media were 50 μg/mL in all the experimental groups, with the control receiving the DMEM media placebo. Before the infection of the Vero cells, *S. typhimurium* strains (both CBD-resistant and CBD-susceptible) were harvested after 6 h of growth and pelleted via centrifugation at their mid-log phase. Bacteria pellets were washed thrice with 1X PBS and resuspended in DMEM media to an OD_600_ of 0.3. To investigate the capacity of the CBD-resistant *S. typhimurium* to resist the killing ability of CBD and thus survive and invade the monolayer Vero cells, the wells containing the infected Vero cells, or the controls, were washed 3 times with 1X PBS after the 30 min treatment with CBD and then stained with acridine red. Bacterial cells stain red in the presence of acridine. Thus, bacterial cells that were able to resist the killing ability of the CBD treatment and hence invaded the Vero cells were captured, whereas *S. typhimurium*, which were susceptible to CBD, died and could not invade the Vero cells, were washed, and thus could not be captured.

### 2.8. LPS Extraction

Gram-negative bacteria are known for their LPS that form a major component of their outer membrane. LPS is critical in resistance to antibiotics, phagocytosis, and serum and serves as an outer membrane permeability barrier. It serves as a major receptor for the majority of phages, e.g., the well-known *Salmonella* P22 and Epsilon 34 phages [30]. In this experiment we extract *S. typhimurium* LPS using the LPS extraction kit (iNtRON Biotechnology, Inc., Dedham, MA, USA), following the manufacturer’s protocol. In brief, *S. typhimurium* cells grown in CBD-tinted LB broth (for the CBD-resistant strain) or LB broth only (for CBD-Susceptible strain) to mid-log phase were centrifuged at 13,226× *g* for 1 min to pellet cells. The bacterial pellet was then washed thrice with 1X PBS to remove all traces of media. Approximately 1 mL of the bacterial lysis Buffer was added to the pellet, and the pellet loosened and was vigorously vortexed. This was then followed by the addition of 200 mL of chloroform, and then vortexed again for 20 s. Then, the sample was incubated for 5 min at 25 °C. Following incubation, the sample was then centrifuged at 13,226× *g* for 10 min at 4 °C, and 400 mL of the supernatant was carefully transferred into a fresh 1.5 mL tube. To the 400 mL of the supernatant, 800 mL of the purification buffer was added and mixed thoroughly, then incubated for 10 min at −20 °C in the freezer. This was then followed by centrifugation at 13,226× *g* for 15 min at 4 °C. The upper layer was then carefully discarded to obtain an LPS pellet. Then, 1 ml of 70% EtOH was added to wash the LPS pellet by inverting the tube 3 times. Then, the mixture was centrifuged for 3 min at 13,226× *g* at 4 °C and the supernatant was discarded to obtain the washed LPS pellet. The LPS pellet was then resuspended in 50 mL of 10 mM Tris-HCl buffer (pH 8.0) and dissolved thoroughly by boiling for 2 min. This was then followed by treatment with a 75 μg/mL concentration of proteinase K at 50 °C for 30 min.

### 2.9. Statistical Analysis

Statistical analyses were performed using OriginPro Plus version 2021b (OriginLab Corporation, Northampton, MA, USA) or Microsoft Excel 2013 (Microsoft, Redmond, WA, USA), and statistical significance was tested via the paired t-test. Immunofluorescence images were analyzed, and image qualities were readjusted by Image J (NIH free open-source software, Version 1.53t). All statistical data were expressed as mean ± Standard error of mean.

## 3. Results

### 3.1. Comparisons of LPS, Ergosterols, Mysristic, Palmitic, and Oleic Acids of Susceptible and Resistant Strains of S. typhimurium

Quantification of the target LPS shows significantly higher abundance of the liposaccharides (Figure 1A) in the CBD-resistant strain as compared to the CBD-susceptible strain. Also, the comparative abundance of the lipids, such as ergosterols (Figure 1B), myristic palmitic (Figure 1C), and oleic acids (Figure 1D), depicts higher abundance in the resistant strain of *S. typhimurium* as compared to the CBD-susceptible strain (Figure 1A–D).

### 3.2. Membrane Fatty Acids Composition of Susceptible and Resistant S. typhimurium

A pie chart showing the membrane sterol compositions of both CBD-resistant and CBD-susceptible strains of *S. typhimurium.* The chart shows that the unsaturated fatty acids were highly abundant in both the CBD-resistant and CBD-susceptible strains, representing 23% and 22%, respectively. The next most abundant membrane fatty acids are the myristic and palmitic acids in both the CBD-resistant and CBD-susceptible strains, representing 28% and 17%, respectively. The least abundant membrane fatty acid being oleic acid, representing 1% in the resistant and 0% in the susceptible strain of *S. typhimurium* (Figure 2).

### 3.3. Immunofluorescence Panels of S. typhimurium Infection of Vero Cells and Treatment with CBD

Immunofluorescence imaging of both CBD-resistant and CBD-susceptible *S. typhimurium* strains treated with CBD. The immunofluorescence studies revealed that the CBD-susceptible strain of *S. typhimurium* was killed by the CBD treatment, as shown in Figure 3, lane 2 (at Texas Red), whereas the CBD-resistant strain of *S. typhimurium* in lane 3 remained alive, thus showing high red fluorescence, hence demonstrating the presence of intracellular *S. typhimurium* in the Vero cells.

### 3.4. Gene Expressions of Susceptible and Resistant Strains of S. typhimurium

To understand the underlying possible genetic drivers that caused the CBD resistance, we investigated some common antibiotic resistance genes such as *int2*, *int3*, *blaTEM*, *fimA*, *fimZ*, *STM0959*, *STM0716*, and the *spy* genes expression in CBD-resistant and CBD-susceptible strains of *S. typhimurium* shown in Figure 4A–C. Comparing the integrons (*int*), *int1* and *int3* were upregulated in the CBD-susceptible strain, while *int2* showed high expression in the CBD-resistant strain as against the CBD-susceptible strain of the *S. typhimurium* (Figure 4A–C).

Similarly, genes coding for bacterial fimbriae (*fimA* and *fimZ*) showed higher expression in the CBD-resistant strain of *S. typhimurium* as compared to the CBD-susceptible strain (Figure 5A,B).

Leucine-response (Lrp) transcriptional regulators with entry number (STM0959) were upregulated in the CBD-resistant strain and downregulated in the CBD-susceptible strain (Figure 6A). Another gene of interest was the DNA recombinase (KEGG Entry number: STM0716), which was downregulated in the CBD-resistant strain; however, in the case of the CBD-susceptible strain, this gene was highly expressed (Figure 6B).

Furthermore, comparing beta-lactamase (*blaTEM*) gene expression in both strains shows that the gene was highly expressed in the CBD-resistant strain as compared to the CBD-susceptible strain of *S. typhimurium* (Figure 7A). Close to 200 relative mRNA expression levels are noted in the *blaTEM* expression in the CBD-resistant strain, whereas negligible expression levels of *blaTEM* were recorded for the CBD-susceptible strain. Similarly, outer membrane protein C (*ompC*) gene expression levels of CBD-resistant strains were depicted to be approximately twice the expression levels of the CBD-susceptible strain (Figure 7B). In the case of the *spy* gene, it showed higher expression in the CBD-resistant strain than in the CBD-susceptible strain of *S. typhimurium* (Figure 7C).

The pie chart below represents a holistic view of all the selected genes considered in this study. The chart shows beta-lactase (*blaTEM*), which occupied more than half of the pie chart, was the highest expressed gene in the CBD-resistant strain of *S. typhimurium* as compared to the other genes. The second highest expression is recorded for *fimZ* in the CBD-resistant strain, followed by STMO716 in the CBD-susceptible strain, STM0959, and *fimA* in the CBD-resistant strain, which also shows significantly high expression profiles (Figure 8).

### 3.5. Interactions of S. typhimurium Resistance Genes with Closely Related Genes

To understand in detail other genes implicated in *S. typhimurium* resistance development, we carried out predictive studies using string network analysis. We determined that other genes closely interact with the selected set of genes under investigation. We found from the prediction analysis that *fimA* is closely interacting with *fimH*, *fimD*, *fimF*, *fimC*, and *fimI* (Figure 9A). The *spy* gene is closely interacting with *cpxA*, *cpxR*, *mdtA*, *STM2535*, *dsbA*, *nlpB*, and *ybaJ* genes (Figure 9B). *ompC* is closely interacting with *yaeT*, *bluB*, *ompA*, *ompX*, *ompR*, and *osoxS* genes (Figure 9C). And *fimZ* is closely interacting with *fimC*, *fimF*, *fimh*, *STM3012*, *torS*, *rcsC*, *umY*, and *ssrB* genes (Figure 9D).

## 4. Discussion

Bacteria’s genetic plasticity has made them remarkable and permits them to adjust to varied environmental stresses imposed on them, including antibiotics that may be deleterious to their existence [42]. Bacteria typically acquire certain intrinsic resistance mechanisms through antimicrobial-producing microorganisms that they share the same ecological niche with, which enables them to withstand antibiotic stresses [43].

In our previous studies, we demonstrated that repeated exposure of *S. typhimurium* to CBD resulted in CBD resistance development [31]. In this study, our primary objective was to ascertain the mechanisms *S. typhimurium* employs to develop resistance to the potent CBD. We discovered that *S*. *typhimurium* CBD-resistance might have arisen from a variety of factors such as LPS and membrane sterols modification and higher expression of specific genetic factors including *blaTEM*, *fimA*, *fimZ*, and *lrp* (SMT0959) genes. LPS is the main structural component of the outer membrane and cell envelope of most Gram-negative bacteria, which is clinically essential due to its role in bacterial pathogenicity and infection [44]. We have observed in this study that there was a distinct difference in the LPS abundance of the CBD-resistant *S. typhimurium* strain as compared to the CBD-susceptible strain (Figure 1A). In this study, we recorded a higher abundance of LPS in the CBD-resistant strain. An abundance of the LPS enhances the outer membrane as a useful permeability barrier for small molecules, as well as hydrophobic molecules that could have crossed the phospholipid bilayers. This property of LPS modification according to Bertani and Ruiz causes significant resistance development of most Gram-negative bacteria to several antimicrobial agents [45]. Cytoplasmic membranes are the defining membranes that separate cells’ internal components from the extracellular matrix and play a crucial role in the transduction of energy and the conservation of solute transport, which regulates cell metabolism [45]. Membranes also function as stress sensors and are capable of transducing signals that might affect genes responsible for defense [45]. Comparing the membrane fatty acids of the CBD-resistant and CBD-susceptible *S. typhimurium* strains reveals some significant and observable differences in unsaturated fatty acids, ergosterols, myristic, palmitic acid, oleic acids, and other membrane sterols as shown in Figure 2. These differences imply that membrane fatty acids play an enormous role in bacteria’s adaptation to CBD. Similar results were obtained by Ayari et al. [45] when they subjected *Bacillus cereus* and *Salmonella typhi* to gamma irradiation treatment to measure the alterations in the membrane fatty acid composition of the two bacteria species. Additionally, OMP protects Gram-negative bacteria from deleterious environments [45]. Simultaneously, several proteins, solutes, and other signal transduction receptors are critical components of the outer membrane, which is critical to the survivability of bacterial cells. OMP acts as a permeability barrier for the exchange of nutrients, passage of toxins, and antibiotics [46]. *ompC* in particular is associated with MRD [47]. Figure 7B shows that the *OmpC* gene was highly expressed in the CBD-resistant *S. typhimurium* strain compared to the CBD-susceptible strain. A study conducted by Liu et al. [48] to understand the role *OmpC* plays in resistance development in *E. coli* by mutating *E. coli* and subsequently treating it with carbapenems and cafepime observed that mutated *E. coli* showed decreased susceptibility to carbapenems and cafepime. In our study, *OmpC* also interacts with other closely related OMPs such as ompA, ompX, and ompR, which are known to be anchor proteins, and play a structural role by maintaining the integrity of the bacteria cell surface. These proteins, especially ompA provide cells with a physical connection between underlying peptidoglycan layers and cell outer membranes (Figure 9C). *OmpC* also interacts with another crucial OMP, such as ompX, which is part of a family of proteins known to be highly virulent and can destabilize the defense mechanism of its host [49]. These related proteins are crucial for the survival of *Salmonella* in macrophages (Figure 9C) [50]. While many pathogens devise new methods of invading host cells, bacterial adhesions serve a crucial function in maximizing viability and survival options. This is achieved by employing type 1 fimbriae, which are markedly varied. *S. typhimurium* produces type 1 fimbriae, which gives them a firm grip on a diverse array of cells and confers them the capacity to withstand stressors [50]. In this study, we considered the expression of the fim genes (*fimA* and fimZ) to elucidate their role in CBD-resistance development. Both *fimA* and *fimZ* were highly expressed in the CBD-resistant strains and were almost absent in the CBD-susceptible strains (Figure 5). Higher *fim* expression allows the bacteria firm attachment and grants it resistibility. Althouse et al. [51] studied the type 1 fimbriae in *S. typhimurium* and concluded that type 1 fimbriae were particularly responsible for *Salmonella* attachment to enterocytes and promoted intestinal colonization. They, however, did not observe any association between type 1 fimbriae and intracellular survival of the *S. typhimurium*. Avalos et al. [51] also studied how type 1 fimbriae help *E. coli* to evade extracellular antibiotics. They found that type 1 fimbriae increased *E. coli* survival chances in macrophages. Additionally, after exposing *E. coli* to gentamicin, they recorded that type 1 fimbriae increased internalization and adhesion to macrophages. The fimbriae are the structural component (main subunit) of the fimbriae. *fimA* and *fimI* are considered adhesive or regulatory genes, and *fimF* is an adapter gene. In this study, *fimA* was highly expressed (Figure 9A). Zeiner et al. [52] assessed *fimA*, *fimF*, and *fimH* to see if they were necessary for assembling type 1 fimbriae in *S. typhimurium*. They achieved this by mutating the *fim* (A, F, and H) genes and examining their capacity to produce surface-assembled fimbriae. Interestingly, they discovered that *S. typhimurium* mutants were unable to assemble fimbriae, which implies that these genes are required to produce fimbriae in *S. typhimurium*. fimZ, together with other related fim genes such as *fimY*, activates fimA proteins, which all play crucial roles in the fimbriae structure assembly (Figure 9D). Avalos et al. [51] investigated the role of *fimW*, *fimY*, and *fimZ* in modulating the expression of type 1 fimbriae in *S. typhimurium*. They indicated that *fimZ* presence could enhance the expression of *hilE*, which is reported to be the repressor of *SP11* gene expression.

Integrons are genetic elements that bacteria employ to withstand and evolve rapidly from stressors through the expression of new genes. The expressed genes are incorporated into a site-specific genetic structure known as gene cassettes that usually carry a non-promoter open reading frame combined with a recombination site. Our study explores various *integron* genes (*int1*, *int2*, and *int3*) to elucidate their role in supporting *S. typhimurium* CBD-resistance development (Figure 4). Only *int2* was shown to be highly expressed in the resistant strains, while both *int1* and *int3* were downregulated in the CBD-resistant strain as compared to the CBD-susceptible strain. Deng et al. [53] critically reviewed the role of integrons (class 1, 2, and 3) in antibiotic resistance by pathogenic bacteria and substantiated that integrons do indeed play a role in resistance development. Jones-Dias et al. [54] studied the architecture of class 1, 2, and 3 integrons from Gram-negative bacteria (*Morganella morganii*, *E*. *coli*, and *Klebsiella pneumoniae*) recovered from fruits and vegetables. They stated that the diverse integrons constituents were associated with transposable elements and led to the identification of varied integron promoters such as *PcW*, *PcS*, *PcH1*, and *PcWTNG-10*. Furthermore, Firoozeh et al. [55] used molecular techniques to characterize class 1, 2, and 3 integrons in clinical MRD-resistant *Klebsiella pneumonia* isolates. They studied the antibiotic resistance patterns of the isolates and found that about 150 isolates carried the *int1* gene, and 55 isolates carried the *int2* gene. *int3* genes were absent in all the isolates. Their finding revealed a strong correlation between *integrons* (especially *int1*) and MRD. Integron-mediated MRD resistance in extended beta-lactamase-producing isolates of *Klebsiella pneumonia* was investigated by MobarakQamsari et al. [56] using 104 clinical isolates. They detected the occurrence of integron classes 1, 2, and 3 in a PCR analysis of the isolates and found that 50 isolates were related to the production of extended-spectrum beta-lactamases. Out of the 50, 22 carried integron classes 1 and 2, and none carried class 3 *integron* genes. They also found that integron-harbored isolates were significantly associated with higher rates of multi-antibiotic (aztreonam, ceftazidime, cefotaxime, cefepime, kanamycin, tobramycin, norfloxacin, and spectinomycin) resistance in extended-spectrum beta-lactamase-producing *Klebsiella pneumonia*. *blaTEM-1 beta-lactamase* gene is the highly expressed gene in the CBD-resistant *S. typhimurium* strain among all the genes considered in this study (Figure 7A and Figure 8). This gene (*blaTEM-1*) is one of the regularly documented antibiotic resistance determinants globally. It is known to exhibit stringent resistance to penicillin and cephalosporins [57]. Several studies demonstrated that *blaTEM-1* plays a crucial role in antibiotic resistance development in some clinically important pathogens such as *Salmonella*, *Neisseria gonorrhoeae*, *E. coli*, *Klebsiella pneumonia*, and *Pseudomonas aeroginosa* [58,59,60]. Yang et al. [61] reported that all their isolates carried the *blaTEM* gene, which was the cause of most of the resistance occurring in *Salmonella* in a large breeder farm in Shandong, China, when they investigated the occurrence and characterization of *Salmonella* isolates from large-scale breeder farms. Similarly, a study carried out in Thailand to determine the antimicrobial genes in *Salmonella* enterica isolates from poultry and swine indicates that the *blaTEM* gene along with other genes such as *cmlA*, *tetA*, *dfrA12*, *sul3*, and *aada1* was detected in majority of the strain [61]. Two variants of the *blaTEM* gene were also prevalent among ampicillin-resistant *E. coli* and *Salmonella* isolated from food animals in Denmark, which were designated *blaTEM-127* and *blaTEM-128*.

The *spy* protein is a chaperone that is expressed and localized in the bacterial periplasm of *E. coli* or *S. typhimurium* during spheroplast formation or by exposure to protein denaturing conditions [62]. Interestingly, *spy* induction has been shown to require regulatory proteins such as BaeR and CpxR, which are part of the major envelope stress response systems BaeS/BaeR and CpxA/CpxR. Thus, scrutinizing *spy* transcription could be a good determinant of the mode of activity of an antimicrobial agent, particularly to observe if it can cause envelope disruption in *E. coli* or in *S. typhimurium* [63].

In this work, we demonstrated for the first time that a CBD-resistant strain of *S. typhimurium* showed higher *spy* expression (Figure 7C); thus, CBD could be used to induce *spy* transcription in *S. typhimurium*. It has been demonstrated previously that agents that induce envelope stresses, including ethanol, polymyxin B, and spheroplasting, cause upregulation in the expression of the *spy* gene in both *S. typhimurium* and *E. coli* [62,63,64]. In this study, the *spy* gene, which is known to be expressed mainly in the periplasmic space under envelope stress conditions, showed significantly higher expression in the resistant strain of *S. typhimurium*, which has been exposed to CBD, than in the susceptible strain, which did not receive CBD treatment. Thus, this work corroborates previous works that have shown that agents that cause envelope stress activate the transcription of *spy*, which then acts as an envelope stress sensor. While previous work in our laboratory demonstrated that CBD caused envelope disruption [28,31], in this work, it is plausible to infer that the CBD-resistant strain of *S. typhimurium*, which showed higher *spy* expression, was due to the induction of the *spy* gene by CBD. It has been demonstrated that the *spy* gene could be used as a whole-cell biosensor for testing antimicrobials that target bacterial cell envelopes [65,66,67]. The *spy* gene interacts with other closely related genes such as *cpxA*, *cpxR*, *mdtA*, *dsbA*, *ybaJ*, *nlpB*, and *STM2535* (Figure 9B). The *nlpB* are nonessential genes that encode lipoproteins and play a crucial role in cell envelope integrity, as was elucidated by Onufryk *et al.* [68]. Jing et al. [69] analyzed the role of *cpxA* mutations in *Salmonella* enterica serovar *typhimurium* resistance to aminoglycosides and β-lactams using site-directed mutants and an internal in-frame mutant and found that *cpxA* and *cpxR* deletion mutants have opposing modulation of resistance to AGAs and β-lactams. Thus, their findings reveal that all *cpxA* mutations increase resistance to AGAs and β-lactams significantly. From this study, it can be safely inferred that these genes, together with the *spy* gene, play a significant role in *S. typhimurium* resistance to CBD. Sterols are an essential part of the lipids of the membranes, and sterols in high concentrations are thought to have beneficial biophysical roles on the membrane. In this study we analyzed the abundance of three different kinds of sterols (ergosterols, myristic, palmitic, palmitoleic, steric, erucic, and oleic acids) in both the CBD-resistant and CBD-susceptible strains of *S. typhimurium* (Figure 1B–D). Sterols are often one of the most abundant biological membranes of eukaryotes and mammals. This study shows a higher abundance of myristic and palmitic acid, ergosterols, and oleic acids in the CBD-resistant *S. typhimurium* strain as compared to the CBD-susceptible *S. typhimurium strain*. The abundance of membrane sterols signifies the ability of the *S. typhimurium* to resist CBD. This is in agreement with findings by Tintino et al. [69], who demonstrated that cholesterol and ergosterols influence the activity of the *Staphylococcus aureus* antibiotic efflux pump.

## 5. Conclusions

In summary, for the first time, we demonstrated that CBD-resistance development in *S. typhimurium* might be caused by several structural and genetic factors. These structural factors were demonstrated here to include LPS and cell membrane sterols, which showed significant differences in abundances on the bacterial cell surfaces between the CBD-resistant and CBD-susceptible strains of *S*. *typhimurium*. Specific key genetic elements implicated in the resistance development investigated included *fimA*, *fimZ*, *int2*, *ompC*, *blaTEM*, DNA recombinase (*STM0716*), leucine-responsive transcriptional regulator (*lrp/STM0959*), and the *spy* gene of *S*. *typhimurium*. In this study, we revealed that *blaTEM* might be the highest contributor to CBD resistance development, indicating the potential gene to target in developing agents against CBD-resistant strains of *S. typhimurium*. To fully understand the mechanisms of resistance development, it will be crucial to investigate all possible genes that might play a role in this event; thus, we are currently employing a follow-up study using throughput RNA sequencing to unravel all the genetic factors blameable in *S*. *typhimurium* development of resistance against CBD. 

## Figures and Tables

**Figure 1 microorganisms-13-00551-f001:**
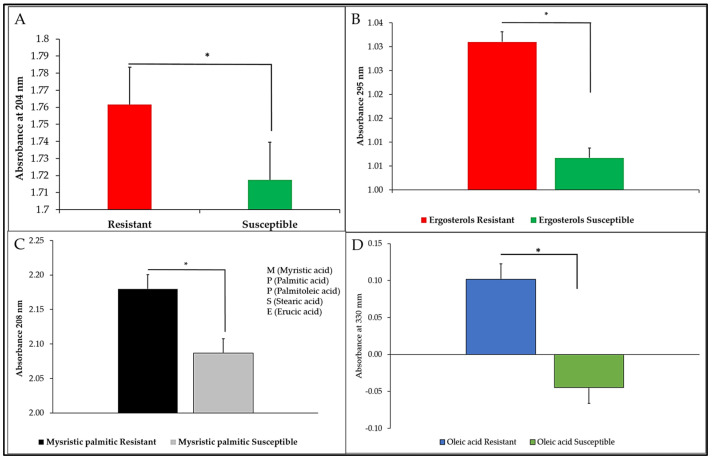
Relative abundance of (**A**) LPS, (**B**) Ergosterols, (**C**) Mysristic palmitic acid, and (**D**) Oleic acid resistance of CBD-susceptible and CBD-resistant *S. typhimurium*. Ergosterol levels (**B**) (*p*-value = 0.389263) show no significant difference between CBD-resistant and CBD-susceptible *S. typhimurium*, suggesting it does not play a major role in resistance. However, palmitic acid (**C**) (*p*-value **=** 0.001483) and oleic acid (**D**) (*p*-value = 0.001687) exhibit highly significant differences, indicating that membrane lipid composition contributes to CBD resistance. Similarly, LPS (**A**) (*p*-value = 0.009726) show significant variation, further supporting the role of membrane lipid alterations in CBD resistance mechanisms. * *p* < 0.05.

**Figure 2 microorganisms-13-00551-f002:**
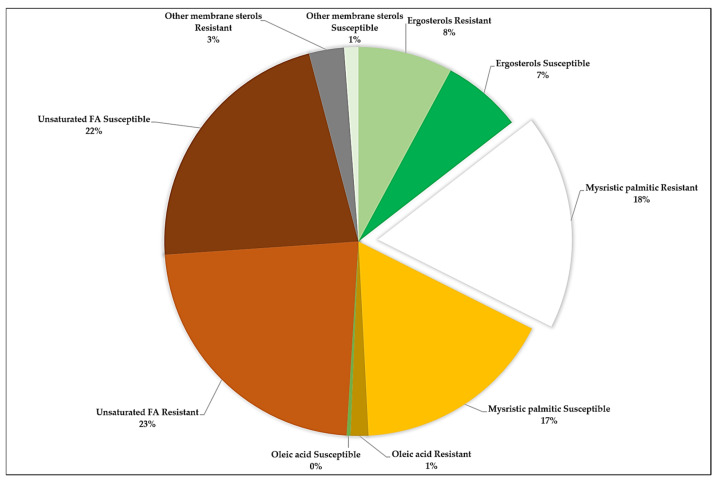
Relative abundance of CBD-Susceptible and CBD-Resistant *S. typhimurium*.

**Figure 3 microorganisms-13-00551-f003:**
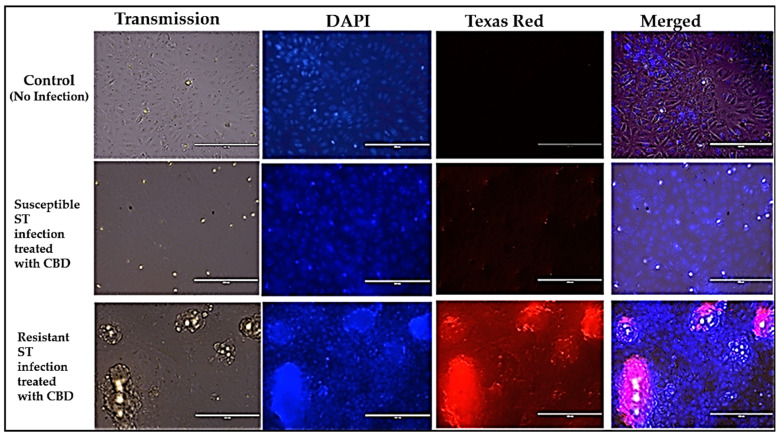
Panels showing the immunofluorescence images of *S. typhimurium* infection of Vero cells and counter treatment with CBD. ST-*S. typhimurium*.

**Figure 4 microorganisms-13-00551-f004:**
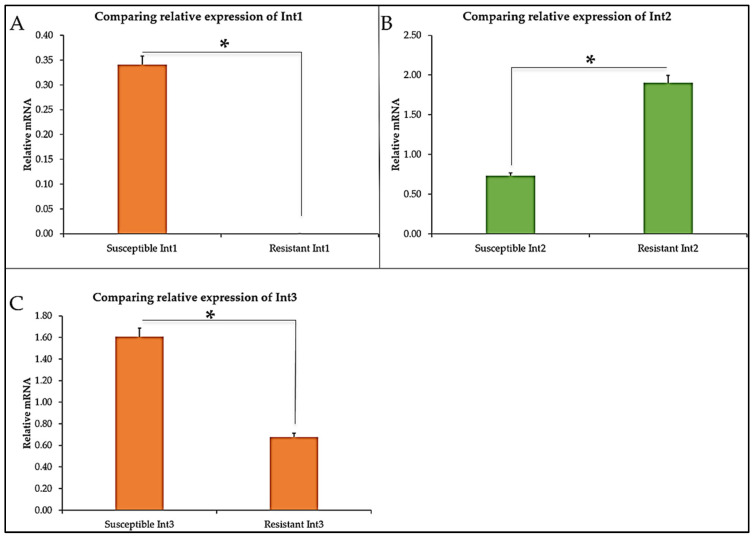
Relative mRNA expression of Integrons in CBD-susceptible and CBD-resistant *S. typhimurium* (**A**) *Integron 1* (**B**) *Integron 2* (**C**) *Integron 3*, * *p*-values < 0.05.

**Figure 5 microorganisms-13-00551-f005:**
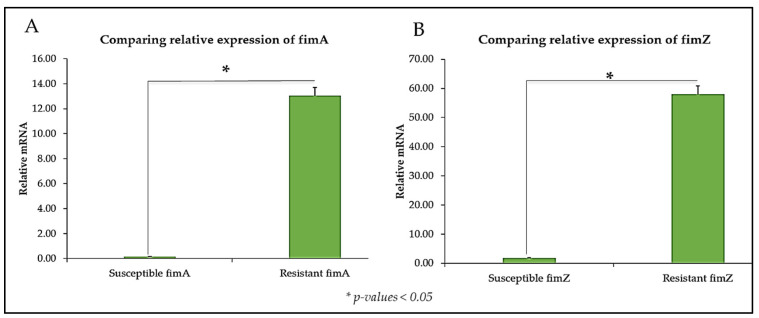
The relative mRNA expression of (**A**) *fimA* and (**B**) *fimZ* in CBD-susceptible and CBD-resistant *S. typhimurium*, * *p*-values < 0.05.

**Figure 6 microorganisms-13-00551-f006:**
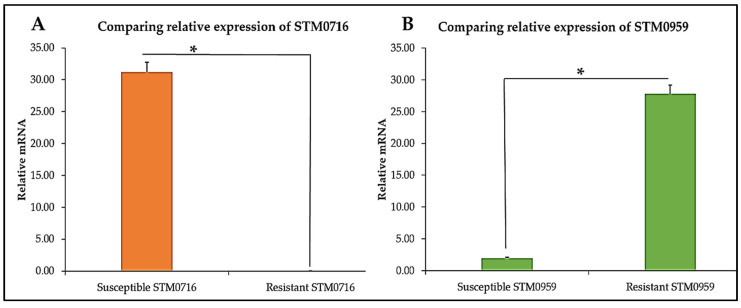
Relative expression of (**A**). *STM0959* and (**B**). *STM0716* in CBD-susceptible and CBD-resistant *S. typhimurium*, * *p*-values < 0.05.

**Figure 7 microorganisms-13-00551-f007:**
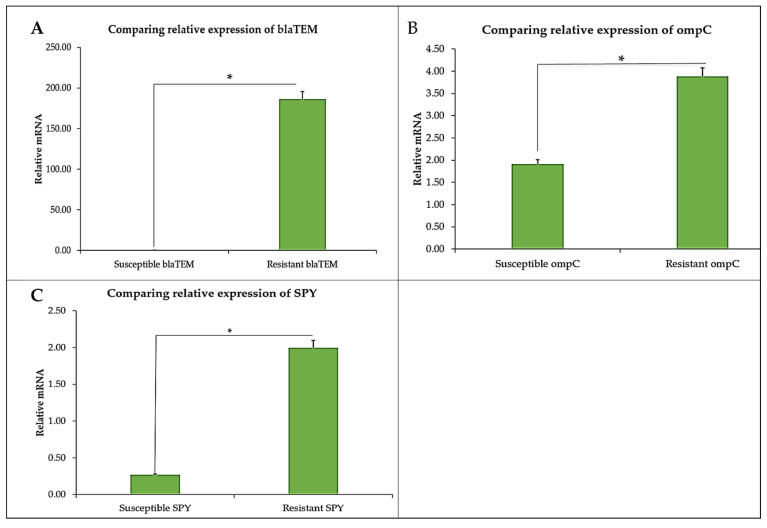
Relative expression of (**A**) *blaTEM*, (**B**) *ompC*, and (**C**) *spy* in CBD-susceptible and CBD-resistant *S. typhimurium*, * *p*-values < 0.05.

**Figure 8 microorganisms-13-00551-f008:**
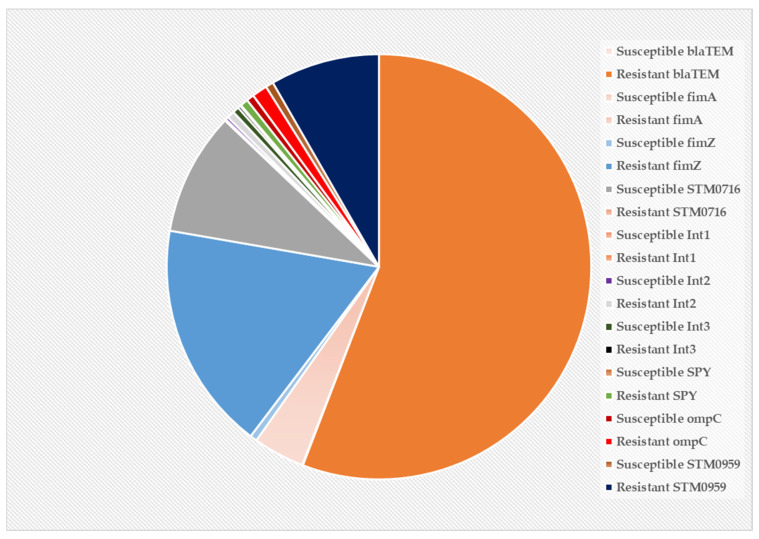
Pie chart of all genes under study that are expressed in CBD-susceptible and CBD-resistant strains of *S. typhimurium*.

**Figure 9 microorganisms-13-00551-f009:**
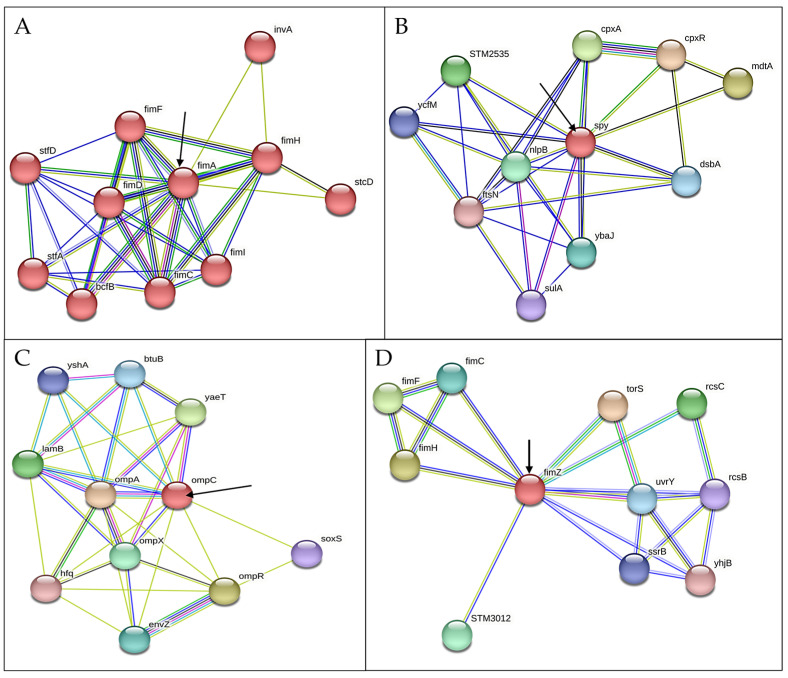
String network depicting Interactions of selected CBD-resistance genes with other closely related genes (**A**) *fimA*, (**B**) *spy*, (**C**) *ompC*, and (**D***) fimZ*.

**Table 1 microorganisms-13-00551-t001:** List of the primers used in the PCR analysis of *int1*, *int2*, *int3*, *blaTEM*, *fimA*, *fimZ*, *STM0959*, *STM0716*, and the *spy* gene expression in CBD-resistant and CBD-susceptible strains of *S. typhimurium*.

Primer	Sequence	Gene Name	Fragment Size/bp	Reference
Fw-*int1*	CCTCCCGCACGATGATC	Integron1	280	[38]
Rv-*int1*	TCCACGCATCGTCAGGC	Integron1	280	[38]
Fw-i*nt2*	TTATTGCTGGGATTAGGC	Integron2	233	[38]
Rv-*int2*	ACGGCTACCCTCTGTTATC	Integron2	233	[38]
Fw-*int3*	AGTGGGTGGCGAATGAGTG	Integron3	600	[38]
Rv-*int3*	TGTTCTTGTATCGGCAGGTG	Integron3	600	[38]
Fw-*ompC*	ATCGCTGACTTATGCAATCG	Outer membrane proteins	204	[39]
RV-*ompC*	CGGGTTGCGTTATAGGTCGT	Outer membrane proteins	204	[39]
Fw*-blaTEM*	ATGAGTATTCAACATTTCCG	Beta-lactamase	859	[40]
Rv-*blaTEM*	ACCAATGCTTAATCAGTGAG	Beta-lactamase	859	[40]
Fw-*fimA*	GCGAGTCTGATGTTTGTCGC	Fimbriae	215	NC_003197.2
Rv-*fimA*	ACGATGGAGAAAGGCACCTG	Fimbriae	215	NC_003197.2
Fw-*fimZ*	GGATGATAGCCGAACAGCGA	Fimbriae	376	NC_003197.2
Rv-*fimZ*	ATAGCGCAGCACGGTAACTT	Fimbriae	376	NC_003197.2
Fw-*STM0716*	CTGTCAGCGACCGACAGAAT	DNA recombinase	115	NC_003197.2
Rv-*STM0716*	CAATATCCGACAAGCGCAGC	DNA recombinase	115	NC_003197.2
Fw-*STM0959*	GCGTATTTCCAACGTCGAGC	leucine-responsive transcriptional regulator	225	NC_003197.2
Rv-*STM0959*	TCTTCAAGCTTTTGCACGGC	leucine-responsive transcriptional regulator	225	NC_003197.2
Fw-*spy*	CGCCAGCGATACCTTCGATA	Spheroplast	205	NC_003197.2
Rw-*spy*	CGCAGCAGGCATTTTACCTT	Spheroplast	205	NC_003197.2
Fw-*gyrB*	GTTGGTGAAGGTTTCGTGGC		458	NC_003197.2
Rv-*gyrB*	ATATCGGCGACACGGATGAC	DNA gyrase subunit B	458	NC_003197.2

## Data Availability

The original contributions presented in the study are included in the article; further inquiries can be directed to the corresponding authors.

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
