# Peer review of "Mechanisms of Salmonella typhimurium Resistance to Cannabidiol"

_microorganisms, 2025, doi:10.3390/microorganisms13030551_

Round 1
Reviewer 1 Report
Comments and Suggestions for Authors
Title: Mechanisms of Salmonella Typhimurium resistance to 2 Cannbidiol.
Manuscript ID: migroorganisms 3465340
I recommended that manuscript could be accepted with MINOR MODIFICATION, based on:
ABSTRACT
- Include quantitative results.
INTRODUCTION
- Chemical formulae of cannbidiol could be positive.
RESULTS
- In figure 1, include mean of ´*´.
- In figure 4, 5, 6 y 7, added statistical analysis.
- Include ´p´ value in the text in each comparison.
GENERAL COMMENTS
- Include IC50/MIC of cannabidiol on susceptible and resistant S. Typhimurium. The author could be selected is will be better in introduction, or as a results that should be described in material and method.
- Cannabidiol don´t need capital letter.
- Review abbreviations use in complete manuscript.
Author Response
Review Report (Reviewer 1)
I recommended that manuscript could be accepted with MINOR MODIFICATION, based on:
ABSTRACT
- Comment 1: Include quantitative results.
Response: Quantitative data included in the abstract
INTRODUCTION
- Comment 2:Chemical formulae of cannbidiol could be positive.
Response: CBD chemical formulae added.
RESULTS
- Comment 3:In figure 1, include mean of ´*´.
Response: P-values added
- Comment 4:In figure 4, 5, 6 y 7, added statistical analysis.
- Response: We have added some statistical analysis to the figures as suggested.
- Comment 5:Include ´p´ value in the text in each comparison.
Response: P-values added
GENERAL COMMENTS
- Comment 6:Include IC50/MIC of cannabidiol on susceptible and resistant S. Typhimurium. The author could be selected is will be better in introduction, or as a results that should be described in material and method.
Response: We have mentioned the MIC values elsewhere in the introduction and also included relevant references where IC50/MIC values relevant to CBD can be obtained.
- Comment 7:Cannabidiol don´t need capital letter.
- Response: Word corrected as suggested.
- Comment 8:Review abbreviations use in complete manuscript.
- Response: Abbreviations reviewed and certified.
Authors Remark: We greatly appreciate your time and the suggestions you gave. Our manuscript looks much better with the integrated suggestions. Thank you.
Reviewer 2 Report
Comments and Suggestions for Authors
In general, the manuscript entitled “Mechanisms of Salmonella Typhimurium resistance to Cannbidiol” is a well-draft manuscript, but the manuscript needs some corrections and editions as follows:
Title
1. “Typhimurium”. Should be in italics.
Abstract
2. Line 12. Replace “multidrug resistance” With “multidrug-resistant (MDR) pathogens”
3. Line 17. Replace “multidrug resistant” With “MDR”
4. L 20-21. “Salmonella Typhimurium (S. Typhimurium)”. Should be in italics.
5. L 23, 25. “S. Typhimurium”. Should be in italics. Apply this comment throughout the manuscript.
6. L26. “blaTEM, fimA, fimZ”. The gene name should be in italics. Apply this comment throughout the manuscript.
7. L28. Add a conclusion sentence.
Keywords
8. The gene name should be in italics.
Introduction
9. L 35-36. Replace “multiple drug resistant (MDR) and extended drug-resistant pathogens (EDRP)” With “multidrug-resistant (MDR) and extensively drug-resistant (XDR) pathogens”. Apply this comment throughout the manuscript.
10. L37. Replace “baumanii” With “baumannii”
11. L39. Replace “Centre for Disease Control and Prevention” With “ Centre for Disease Control and Prevention (CDC)”
12. L50. Replace “United States” With “United States (US)”.
13. L51. “salmonellosis,” Not italic. Apply this comment throughout the manuscript.
14. L 64,66,68,70. Replace “Cannabidiol (CBD)” With “CBD”. Apply this comment throughout the manuscript.
15. L72. L50. Replace “United States” With “US”. Apply this comment throughout the manuscript.
16. L 79. “Streptococci and Staphylococci” The name of the bacteria should be in italics. Apply this comment throughout the manuscript.
17. L85. Replace “Typhimurium and Newington” With “S. Typhimurium and S. Newington”
18. L 88. “S. Typhimurium”. Should be in italics. Apply this comment throughout the manuscript.
Materials and Methods
19. L 102. “LB agar”. State the full name followed by the abbreviation in brackets, then state the name of the company, city, and country from which you obtained the used media.
20. 104-106. “Luria Broth (BD, Difco, Franklin Lakes, 104 NJ, USA) and Luria Agar (BD, Difco, Franklin Lakes, NJ, USA) were the media used for culturing and maintenance of the bacteria.” Move this sentence to line 100.
21. L110. Replace “CO2”. With “CO2”
22. L119. “PBS”. State the full name followed by the abbreviation in brackets.
23. L124. Convert rpm to g force. Apply this comment throughout the manuscript.
24. L 170-177. The gene name should be in italics. Apply this comment throughout the manuscript.
25. L 178. Table 1. Add a column indicating the gene name.
26. L183. Replace“105” With “105”. Please revise all superscript and subscript letters and numbers throughout the manuscript.
Results
27. L236. Replace “liposaccharides (LPS)” With “LPS”.
28. L361. Replace “outer membrane proteins (OMP)” With “OMP”.
29. L334-406. This paragraph is too long try to split it into smaller paragraphs.
30. L245. “p”. Should be in italics. Apply this comment throughout the manuscript.
Discussion
31. L 339. Replace “Lipopolysaccharide (LPS)” With “Lipopolysaccharide”
Author Response
Review Report (Reviewer 2)
Comments and Suggestions for Authors
In general, the manuscript entitled “Mechanisms of Salmonella Typhimurium resistance to Cannbidiol” is a well-draft manuscript, but the manuscript needs some corrections and editions as follows:
Title
Comment 1. “Typhimurium”. Should be in italics.
Response: Name corrected as pointed out.
Abstract
Comment 2. Line 12. Replace “multidrug resistance” With “multidrug-resistant (MDR) pathogens”
Response: Changes have been effected as pointed out.
Comment 3. Line 17. Replace “multidrug resistant” With “MDR”
Response: Changes made as pointed out.
Comment 4. L 20-21. “Salmonella Typhimurium (S. Typhimurium)”. Should be in italics.
Response: Name corrected as suggested.
Comment 5. L 23, 25. “S. Typhimurium”. Should be in italics. Apply this comment throughout the manuscript.
Response: Names corrected throughout the entire manuscript as suggested.
Comment 6. L26. “blaTEM, fimA, fimZ”. The gene name should be in italics. Apply this comment throughout the manuscript.
Response: Names corrected throughout the entire manuscript as suggested.
Comment 7 L28. Add a conclusion sentence.
Response: The section is rewritten to include a concluding sentence.
Keywords
Comment 8 The gene name should be in italics.
Response: Names corrected throughout the entire manuscript as suggested.
Introduction
Comment 9. L 35-36. Replace “multiple drug resistant (MDR) and extended drug-resistant pathogens (EDRP)” With “multidrug-resistant (MDR) and extensively drug-resistant (XDR) pathogens”. Apply this comment throughout the manuscript.
Response: Names corrected throughout the entire manuscript as suggested.
Comment 10. L37. Replace “baumanii” With “baumannii”
Response: Names corrected as suggested.
Comment 11. . L39. Replace “Centre for Disease Control and Prevention” With “ Centre for Disease Control and Prevention (CDC)”
Response: Names corrected as suggested.
Comment 12. L50. Replace “United States” With “United States (US)”.
Response: Names corrected as suggested.
Comment 13. L51. “salmonellosis,” Not italic. Apply this comment throughout the manuscript.
Response: Names corrected as suggested.
Comment 14. L 64,66,68,70. Replace “Cannabidiol (CBD)” With “CBD”. Apply this comment throughout the manuscript.
Response: Names corrected as suggested.
Comment 15. L72. L50. Replace “United States” With “US”. Apply this comment throughout the manuscript.
Response: Names corrected as suggested.
Comment 16. L 79. “Streptococci and Staphylococci” The name of the bacteria should be in italics. Apply this comment throughout the manuscript.
Response: Names corrected as suggested.
Comment 17. L85. Replace “Typhimurium and Newington” With “S. Typhimurium and S. Newington”
Response: Names corrected as suggested.
Comment 18. L 88. “S. Typhimurium”. Should be in italics. Apply this comment throughout the manuscript.
Response: Names corrected as suggested throughout the entire manuscript.
Materials and Methods
Comment 19. L 102. “LB agar”. State the full name followed by the abbreviation in brackets, then state the name of the company, city, and country from which you obtained the used media.
Response: We have provided the full details for L.B agar as suggested.
Comment 20. 104-106. “Luria Broth (BD, Difco, Franklin Lakes, 104 NJ, USA) and Luria Agar (BD, Difco, Franklin Lakes, NJ, USA) were the media used for culturing and maintenance of the bacteria.” Move this sentence to line 100.
Response: Sentence have been moved to the right line as suggested.
Comment 21. L110. Replace “CO2”. With “CO2”
Response: Names corrected as suggested.
Comment 22. L119. “PBS”. State the full name followed by the abbreviation in brackets.
Response: Names corrected as suggested.
Comment 23. L124. Convert rpm to g force. Apply this comment throughout the manuscript.
Response: rpm values converted to g force.
Comment 24. L 170-177. The gene name should be in italics. Apply this comment throughout the manuscript.
Response: Names corrected as suggested.
Comment 25. L 178. Table 1. Add a column indicating the gene name.
Response: we have added column with the corresponding names to each gene used.
Comment 26.. L183. Replace“105” With “105”. Please revise all superscript and subscript letters and numbers throughout the manuscript.
Response: Number corrected as suggested.
Results
Comment 27. L236. Replace “liposaccharides (LPS)” With “LPS”.
Response: Names corrected as suggested.
Comment 28. L361. Replace “outer membrane proteins (OMP)” With “OMP”.
Response: Names corrected as suggested.
Comment 29. L334-406. This paragraph is too long try to split it into smaller paragraphs.
Response: Here we cannot relate to the lines you are referring to.
Comment 30. L245. “p”. Should be in italics. Apply this comment throughout the manuscript.
Response: Names corrected as suggested.
Discussion
Comment 31. L 339. Replace “Lipopolysaccharide (LPS)” With “Lipopolysaccharide”
Response: Names corrected as suggested.
Authors Remark: We greatly appreciate your time and the suggestions you gave. Our manuscript looks much better with the integrated suggestions. Thank you.

Reviewer 3 Report
Comments and Suggestions for Authors
The manuscript is fine overall, I do have some suggestions and comments:
- Line 97 CBDSusceptible - CBD-Susceptible
- The strain of S. Typhimurium used in this study - do we have any information about it's susceptibility to antibiotics? Also, the authors mention it is their laboratory strain BV4012 but refer to it as "S. Typhimurium LT2 98 strain MS1868", which the reviewer finds confusing. If it is unmodified MS1868 please use this name of the strain as it has been characterized and published. If it is a MS1868-derived strain please state it clearly in the paper
- the design of the figures is fine the letters are too small. It is impossible to read what is what on the graphs without enlarging the images
- Line 367 S. typhimurium - S. Typhimurium
- E. coli in Discussion should be in italics
- Read throught the text of the manuscript and correct the names of genes and proteins. Gene names should be in italics, while protein names start with a capital letter. The reviewer found at least few instances when the genes are written without the use of italics.
Author Response
Author's Reply to the Review Report (Reviewer 3)
Comments and Suggestions for Authors
The manuscript is fine overall, I do have some suggestions and comments:
Comment 1: Line 97 CBDSusceptible - CBD-Susceptible
Response: Names corrected as suggested.
Comment 2. The strain of S. Typhimurium used in this study - do we have any information about it's susceptibility to antibiotics? Also, the authors mention it is their laboratory strain BV4012 but refer to it as "S. Typhimurium LT2 98 strain MS1868", which the reviewer finds confusing. If it is unmodified MS1868 please use this name of the strain as it has been characterized and published. If it is a MS1868-derived strain please state it clearly in the paper
Response: we have corrected the statement to correct show that the CBD-resistant strain was created in our lab from the CBD-susceptible strain (BV4012).
Comment 3: the design of the figures is fine the letters are too small. It is impossible to read what is what on the graphs without enlarging the images
Response: We have slightly readjusted the figures for better visuals.
Comment 4: Line 367 S. typhimurium - S. Typhimurium
Response: We have corrected the name as pointed out.
Comment 4: E. coli in Discussion should be in italics
Response: Name corrected as pointed out.
Comment 5: Read throught the text of the manuscript and correct the names of genes and proteins. Gene names should be in italics, while protein names start with a capital letter. The reviewer found at least few instances when the genes are written without the use of italics.
Response: We have gone through the entire manuscript to ensure that all genes and protein names are corrected as pointed out.
Authors Remark: We greatly appreciate your time and the suggestions you gave. Our manuscript looks much better with the integrated suggestions. Thank you.